# Factors Affecting the Number of Pollen Grains per Male Strobilus in Japanese Cedar (*Cryptomeria japonica*)

**DOI:** 10.3390/plants10050856

**Published:** 2021-04-23

**Authors:** Hiroyuki Kakui, Eriko Tsurisaki, Rei Shibata, Yoshinari Moriguchi

**Affiliations:** 1Graduate School of Science and Technology, Niigata University, Niigata City, Niigata 950-2181, Japan; kakui@agr.niigata-u.ac.jp; 2Faculty of Agriculture, Niigata University, Niigata City, Niigata 950-2181, Japan; f21e086c@mail.cc.niigata-u.ac.jp (E.T.); rshibata@agr.niigata-u.ac.jp (R.S.)

**Keywords:** pollen, pollinosis, cell counter, genetic variation, piecewise Structural Equation Modeling (pSEM), model prediction

## Abstract

Japanese cedar (*Cryptomeria japonica*) is the most important timber species in Japan; however, its pollen is the primary cause of pollinosis in Japan. The total number of pollen grains produced by a single tree is determined by the number of male strobili (male flowers) and the number of pollen grains per male strobilus. While the number of male strobili is a visible and well-investigated trait, little is known about the number of pollen grains per male strobilus. We hypothesized that genetic and environmental factors affect the pollen number per male strobilus and explored the factors that affect pollen production and genetic variation among clones. We counted pollen numbers of 523 male strobili from 26 clones using a cell counter method that we recently developed. Piecewise Structural Equation Modeling (pSEM) revealed that the pollen number is mostly affected by genetic variation, male strobilus weight, and pollen size. Although we collected samples from locations with different environmental conditions, statistical modeling succeeded in predicting pollen numbers for different clones sampled from branches facing different directions. Comparison of predicted pollen numbers revealed that they varied >3-fold among the 26 clones. The determination of the factors affecting pollen number and a precise evaluation of genetic variation will contribute to breeding strategies to counter pollinosis. Furthermore, the combination of our efficient counting method and statistical modeling will provide a powerful tool not only for Japanese cedar but also for other plant species.

## 1. Introduction

Pollen grain number is an important trait in plants. From an evolutionary perspective, the number of pollen grains produced has a direct effect on the number of offspring, and is known to vary among mating systems. For example, outcrossing plants tend to produce more pollen grains compared to closely related selfing species [1,2,3,4]. Among outcrossing plants, wind-pollinators tend to produce more pollen grains than animal-pollinated species [5,6]. From an agricultural perspective, while domesticated crop species such as rice or wheat tend to produce fewer pollen grains, effective pollination with abundant pollen grains is sometimes needed for purposes such as breeding hybrid crops or artificial pollination [7,8,9,10,11,12]. Pollen is also important from a medical standpoint, because it triggers an allergic reaction called pollinosis around the world [13,14].

Species in the family Cupressaceae are distributed worldwide [15]. While the family includes many economically-valuable species used for timber or ornamental purposes, some species, such as cypress (*Chamaecyparis obtusa*), mountain cedar (*Juniperus ashei*), and Japanese cedar (*Cryptomeria japonica*) have been linked to serious pollinosis [16]. Japanese cedar is the primary source of building materials in Japan, and comprises ~43% of plantation stands (~4.4 million ha) [17]. Trees with more desirable traits (e.g., timber quality, growth rates, disease tolerance, environmental tolerances, etc.) have been selected and propagated using cuttings or seedlings [18]. However, the pollen of Japanese cedar is the primary cause of pollinosis in Japan, and in 2008, this species caused pollinosis in 26.5% of the population, a drastic increase from 16.2% in 1998 [19]. Moreover, Japanese cedar pollinosis tends to be more serious in Tokyo: the Bureau of Social Welfare and Public Health of Tokyo Metropolitan Government recently reported that 45.6% of patients with pollinosis from Japanese cedar were in Tokyo [20]. As such, pollen production per tree is among the most important considerations in breeding strategies for Japanese cedar. Generally, the total number of pollen grains produced by a single tree is the product of the number of pollen grains per male strobilus (male flower) and the number of male strobili per tree. For phenotyping, it is relatively simple to quantify the number of male strobili, as they are macroscopic and can be easily counted. More than 140 clones of Japanese cedar are classified as low pollen clones [21]; however, these were defined only by the number of male strobili, and pollen production per male strobilus was not considered. The number of pollen grains per male strobilus also affects the total pollen production of a single tree, but little is known about this trait as it is not visible to the naked eye, and methods for counting pollen grains are not well-established.

Microscopes have conventionally been used for counting pollen grains, and have been used to quantify pollen production in Japanese cedar [22,23,24]. However, this approach may be time consuming and/or inefficient. For example, the microscope method typically requires approximately 17 min to count ~200 pollen grains [25]. Recently, we developed a pollen counting method for Japanese cedar that uses mesh columns and a cell counter [26]. The cell counter method can count approximately 20,000 particles per measurement within five minutes (i.e., >3 times faster and counts > 100 times the number of particles).

Here, we focus on pollen grain production per male strobilus (hereafter “pollen number”). We counted pollen numbers from 26 clones of Japanese cedar and used piecewise Structural Equation Modeling (pSEM) to identify factors that affect pollen number. pSEM, which has gained attention in recent years in the fields of ecology and evolution, can reveal complex relationships among various factors by constructing a path diagram [27,28,29,30,31]. Because trees grow densely in breeding stands, it can be difficult to sample male strobili from branches of the same height and facing the same direction. Thus, we collected male strobili representing different directions and heights. We used statistical modeling to obtain a corrected pollen number that controls for the effects of these differences in environmental conditions, thus facilitating comparisons among clones.

## 2. Materials and Methods

### 2.1. Plant Materials and Sampling Conditions

We collected 523 male strobili from 26 Japanese cedar clones in the Niigata Prefectural Forest Research Institute (38°16′22.7′′ N 139°30′35.4′′ E). We collected male strobilus samples in February 2020 because the pollen grains are developmentally fully mature but the male strobilus has not yet opened at this time [32]. All data are shown in Appendix A and summary data are provided in Table 1. Male strobili were always collected from the open-air side of the tree (the side where sunlight is not blocked by other trees; Figure 1a,c) because sunlight positively affects the growth of male strobili [33]. Male strobili from Wogon-sugi and Iwafune-15 were collected from multiple directions (Table 1, Figure 1c), and male strobili from Wogon-sugi, Iwafune-2, and Iwafune-16 were collected at different heights (above or below 5 m; Table 1, Figure 1b,d). At least two inflorescences were collected per clone (Figure 1e,f, Table 1). Pollen number was estimated by counting at least six male strobili per clone (Table 1).

### 2.2. Male Strobili Measurement

Male strobili were weighed using a digital precision scale (ViBRA-HT, Shinko Denshi, Tokyo, Japan). Images of samples were taken under a stereomicroscope (SZ-11, Olympus, Japan) using a digital camera (WRAYCAM-EL310, WRAYMER, Osaka, Japan). Pollen grain size was measured using ImageJ and Fiji software [34]. Representative photos from small, medium, and large male strobili are shown in Figure 2a–c, respectively (also see Section 3.1).

### 2.3. Measurement of Pollen Number and Size Using a Cell Counter

Pollen number and size were measured as previously described [26]. Briefly, a single male strobilus was gently crushed with a pestle and suspended in water. The pollen suspension was then filtered through two mesh filters (100 and 20 μm) to remove debris and small particles. The cleaned suspension was mixed with an isotonic cell counter buffer, and pollen number and pollen size (diameter) were assessed using a cell counter (CASY cell counter, OMNI Life Science, Bremen, Germany). Representative results for the samples shown in Figure 2a–c are presented in Figure 2d–f. Data were analyzed using the CASY application (OMNI Life Science) and plots were constructed in R [35].

### 2.4. Evaluation for the Hypothesized and Best Fit Pathways Using Statistical Modeling (pSEM)

We used pSEM to examine the direct and indirect effects of environmental conditions and other factors on pollen number. First, we constructed a full pathway model, shown in Figure 3a, which included all hypothesized causal relationships. We then fitted a linear mixed effect model (LMM) for each of three response variables; male strobilus weight and pollen size were modeled as a function of direction and height, whereas pollen number was modeled as a function of male strobilus weight and pollen size. Direction was converted into a numerical variable in which −1 represented north, −0.5 represented northwest and northeast, 0 represented west and east, 0.5 represented southwest and southeast, and 1 represented south. Collection height was converted to a dummy variable with 0 indicating collection below 5 m and 1 indicating collection above 5 m. Before fitting the LMMs, male strobilus weight and pollen number were log-transformed to meet assumptions of normality. We included genetic contributions (i.e., variations associated with clone) as random effects in the LMMs as these clearly affected response variables. Inflorescence was also included as a random effect to account for the bias associated with the nested sampling structure (i.e., the effects of collecting multiple samples from a single inflorescence), but this is not discussed further as it was not our focus. Male strobilus weight and pollen size were clearly correlated, but the direction of causality was not clear; these two variables were therefore treated as correlated errors. The three abovementioned LMMs and the two correlated errors were integrated into the full hypothesized pathway model (hypothetical model). We then simplified the hypothetical model by removing less significant paths to improve overall goodness of fit. Goodness of fit was compared among models using the Akaike information criterion (AIC), and the removal of a path was accepted if it decreased the AIC value [36]. We selected the pSEM with the lowest AIC value as the best fit model (best model). For both the hypothetical model and best model, we extracted the standardized effect size of all paths to determine path strength. We also calculated the variance (*R*^2^) explained by each predictor and by genetic contributions, to compare their individual importance in the LMMs [37]. The pSEM models and LMMs were constructed using the *piecewiseSEM* and *lmerTest* R packages, respectively [27,38].

### 2.5. Corrected Pollen Number by Statistical Modeling (LMM)

Using the three LMMs included in the best model, we predicted the expected pollen number for each clone after correcting for the effects of sampling direction. The effect of sampling height was disregarded because it was not a component of the best model (Figure 3b). First, we extracted the random effects of clone, which represent within-group variation among the response variables (i.e., male strobilus weight, pollen size, and pollen number), from the LMMs. We then extracted the predicted mean and 95% confidence intervals (CIs) for male strobilus weight and pollen size for each clone, assuming that the effects of sampling direction on response variables were zero, i.e., that sampling direction was east or west. Finally, we extracted the predicted mean pollen number and 95% CIs for each clone, assuming that values of the predictors (i.e., male strobilus weight and pollen size) were the predicted mean ± 95% CIs of each clone predicted above.

## 3. Results and Discussion

### 3.1. Male Strobilus Characteristics and Genetic/Environmental Conditions

We assessed a total of 523 male strobili collected from 26 clones. Male strobili were weighed and photographed before pollen was counted. Male strobili varied considerably in weight and size (Appendix A, Figure 2a–c). Male strobilus weight and area were strongly correlated (Appendix A, r = 0.892). Hereafter we discuss only male strobilus weight, as it is a trait that can be easily determined and it is considered more precise than area (area only contains information on two dimensions of the male strobilus, whereas weight includes information about the entire sample). Cell counter results indicated that larger/heavier male strobili tended to have more pollen grains (Figure 2), and that pollen size varied among samples (Figure 2d–g). We analyze and discuss the factors affecting pollen number in more detail in the following sections.

### 3.2. Hypothetical and Best Models

To explore the factors affecting pollen number, we first specified five input conditions: pollen number per male strobilus (“pollen number”); pollen grain diameter (“pollen size”); male strobilus weight (“male strobilus weight”); sampling direction (“direction”); and sampling height (“height”). We also incorporated a factor (“clone”) to account for the genetic variation. The hypothetical model was constructed using pSEM, and is shown in Figure 3a. In this hypothetical model, direction had a significant effect on pollen number, whereas the effects of height were non-significant. This result was unexpected as we had initially assumed that both direction and height would affect the growth of male strobili due to their effects on light received. Our findings indicate that direction is more significant than height because we always collected samples from the open-air side of the tree (Figure 1a–d). In the open-air, the total amount of sunlight may not differ by height, but the amount of sunlight per year clearly differs by direction [39]. Although direction significantly affected male strobilus weight, effects on pollen size were non-significant (Figure 3a). We then constructed the best model (Figure 3b) by simplifying the hypothetical model. Comparison of AIC values indicated that the best model was better than the hypothetical model (Figure 3). The best model revealed that pollen number is explained well by male strobilus weight, clone, and pollen size (total contribution 85.2%). Effect size (black arrow) revealed that male strobilus weight positively affects the pollen number, while pollen size negatively affects the pollen number. Male strobilus weight is affected by a genetic contribution (clone) and by direction as an environmental condition. Pollen size is mainly affected by the genetic contribution, and mutually affects male strobilus weight. Overall, pSEM revealed which factors significantly affect other factors and successfully determined the contributions. We analyze and discuss each factor in more detail in the following sections.

### 3.3. Factors Affecting Pollen Number

The best model indicated that pollen number was significantly affected by male strobilus weight and pollen size, and was influenced by genetic contributions (Figure 3b). Figure 4a presents a bar plot of pollen number by clone; pollen number varied from approximately 100,000 to 300,000, indicating that genetic contributions substantially influenced pollen number, as indicated by the pSEM (Figure 3b). Figure 4b presents the model of pollen number using all samples and two predictors (male strobilus weight and pollen size). Male strobilus weight was strongly and positively correlated with pollen number, as suggested by the model (Figure 3b). Every 10-mg increase in male strobilus weight was associated with an increase in pollen number of approximately 100,000 (Figure 4b). Moreover, for male strobili of equal weight, samples with a smaller pollen size tended to have a higher pollen number (Figure 4b, light blue vs. orange symbols). This finding suggests that pollen size also affects the pollen number. Kondo et al. (1992) counted pollen grains in 10 clones and they also found a correlation between male strobilus weight and pollen number [23]. We propose that pollen number can be estimated more accurately using male strobilus weight and pollen size.

### 3.4. Factors Affecting Male Strobilus Weight

pSEM and pollen number modeling suggested that male strobilus weight is among the most important factors influencing pollen number. pSEM predicted that male strobilus weight is significantly affected by direction and genetic contributions (Figure 3b and Figure 5a). A lower pollen number was associated with lower male strobilus weight, and vice versa (Figure 4a and Figure 5a); however, several clones, including Higashikubiki-5, Iwafune-17, Nishikanbara-1, and Iwafune-15, exhibited only weak relationships between pollen number and male strobilus weight. Higashikubiki-5 and Nishikanbara-1 generally had low male strobilus weights but high pollen numbers, while the inverse was true for Iwafune-17 and Iwafune-15 (Figure 4a and Figure 5a). This apparent contradiction was resolved by accounting for pollen size: Higashikubiki-5 and Nishikanbara-1 had smaller pollen grains whereas Iwafune-17 and Iwafune-15 had larger grains (Figure 6a). These results support the hypothesis that pollen number is affected by not only male strobilus weight but also by pollen size.

To explore the relationship between male strobilus weight and direction, we generated violin plots based either on all 523 samples, or the 101 samples collected from Wogon-sugi, which were collected from all directions (Figure 5b,c). Both datasets suggest that male strobili collected from the south side of trees tended to have approximately 1.5-fold greater male strobilus weight compared to those collected from the north side (1.475-fold for all data [Figure 5b], 1.458-fold for the Wogon-sugi subset [Figure 5c]). This is likely due to the amount of sunlight that male strobili are exposed to, as the south sides of trees in Japan receive more sunlight than the north sides [39]. It is known that the amount of solar radiation positively affects the male strobilus number [33]. Our data also indicate that the amount of sunlight also positively affects male strobilus weight (pollen number indirectly).

### 3.5. Factors Affecting Pollen Size

Few studies have examined the pollen size variation among Japanese cedar clones. The pSEM predicted that pollen size is significantly correlated with male strobilus weight and influenced by genetic variation (Figure 3b). Plot data showed that pollen size varied from 32 to 36 μm among clones (Figure 6a). Interestingly, the mean rank of pollen size was not significantly correlated with the mean rank of pollen number (Spearman’s rank correlation = 0.212, *p* = 0.298; Figure 4a) or mean male strobilus weight (Spearman’s rank correlation = 0.305, *p* = 0.129; Figure 5a). This suggests that pollen size follows a pathway that is independent from pollen number. Figure 6b shows the relationship between male strobilus weight, pollen size, and clone; samples from different clones are represented by different symbols Table 1. We draw two conclusions from this plot; first, samples from the same clone were clustered, and second, within the same clone, heavier male strobili tended to be associated with large pollen. These results suggest that pollen size is primarily controlled by genetic variation, but is also affected by male strobilus weight. Our results suggest that male strobilus weight is also positively related to sunlight, and, as such, that sunlight indirectly influences pollen size.

### 3.6. Direction-Corrected Pollen Number Allows for More Accurate Comparisons of Pollen Number among Clones

It is often difficult in field studies to sample in a way that is not confounded by variations in environmental conditions. We collected male strobilus samples from branches facing different directions because the open-air side of the tree differed among clones (Table 1, Figure 1a,c). Ideally, comparisons of pollen numbers among clones should control for the effects of variations in environmental conditions, in our case direction, as we have demonstrated that pollen number is indirectly affected by direction via variations in male strobilus weight (Figure 3b and Figure 5b). We corrected the pollen number for each clone by constructing LMMs that included direction as a predictor, and extracted the mean pollen number per clone for samples representing the same direction. Raw and corrected pollen numbers are shown in Figure 7. In some cases, the corrected pollen numbers differed from the raw data (e.g., Minamikanbara-3, Iwafune-15, Iwafune-1, and Iwafune-12). Corrected pollen numbers enable us to make more precise comparisons among clones.

Our data indicate that the mean pollen number varied approximately 3-fold among clones, from 100,000 to 300,000 pollen grains per male strobilus. Three studies have calculated pollen grain number per male strobilus: Ikuse (1965) reported that single male strobilus produces 396,000 pollen grains [22]; Saito and Takeoka (1987) found 237,000 to 442,000 pollen grains from six clones [24]; and Kondo et al. (1992) reported from 182,019 to 656,289 pollen grains in 10 clones [23]. We found that clones produced from 100,000 to 300,000 pollen grains each; fewer than the other reports. There are several possible reasons for this, including genetic variation. We found that pollen number has a large genetic variation among the clones and we did not examine the same clones studied in the other reports. In *C. japonica*, DNA markers indicate that there is genetic differentiation between clones growing mostly on the Pacific Ocean side of Japan and those growing mostly on the Japan Sea side [40]. We measured clones from Niigata Prefecture, on the Japan Sea side, while the previous studies measured many clones from the Pacific Ocean side [22,23,24]. Additionally, because it is known that male strobilus number varies markedly by year, annual differences in the pollen number per male strobilus are possible [33,41]. Counting pollen grains from a larger area over multiple years will provide more comprehensive data on pollen numbers.

To counter pollinosis, pollen number per male strobilus can possibly contribute some aspects. For example, we can further select a clone with a low pollen number from the clones that have selected by few numbers of male strobilus. It is important to consider the amount of allergen to assess its effects on pollinosis patients. Cry j 1 is a major Japanese cedar allergen and the amount of Cry j 1 (μg g^−1^) differs up to 80 times difference among the clones [42]. One interesting approach would be to analyze the correlation between the amount of allergen and pollen number (or pollen size). Introducing the pollen number to a Japanese cedar pollinosis breeding strategy may help counter pollinosis.

Pollen grain number varies in several plant species and the locus or gene controlling pollen number was discovered recently. Nguyen et al. investigated rye chromosome addition lines in wheat and found that the 4R addition line had 33% more pollen, which might have applications in breeding hybrid wheat [43]. Recently, the gene controlling pollen numbers was identified in *Arabidopsis thaliana* [44]; the pollen number varied around 4-fold among 144 *Arabidopsis thaliana* accessions (approximately 2000–8000 pollen grains per flower). Functional analysis revealed *REDUCED POLLEN NUMBER 1 (RDP1)* as the gene controlling pollen number, and an evolutional analysis revealed that *RDP1* is related to the recent pollen reduction in selfing *A. thaliana*. Although Japanese cedar has much larger genome (10.8 GB) than *A. thaliana*, a recent genetic study identified the male sterility gene in Japanese cedar [45,46]. Combination with DNA marker analysis and efficient pollen counting method may contribute to revealing the pollen number controlling locus (or gene) in the future.

## 4. Conclusions

We quantified the number of pollen grains per male strobilus using an efficient counting method, complemented by modeling and statistical methods. Modeling revealed which factors affect pollen numbers in Japanese cedar, and corrected pollen numbers enabled us to make accurate comparisons among clones. Because we found more than 3-fold variation among clones, counting pollen grain numbers will help to select clones with less pollen. Modeled predictions will also be useful for examining other traits in the field research.

## Figures and Tables

**Figure 1 plants-10-00856-f001:**
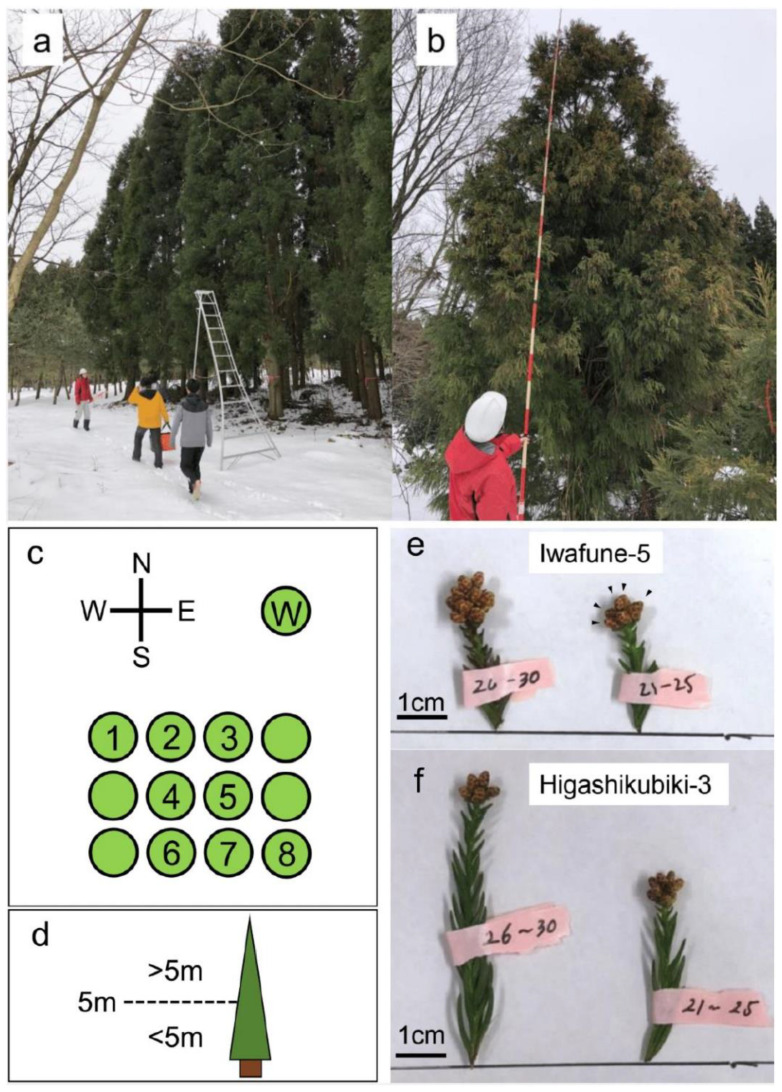
Sample collection. Male strobilus samples were always collected from the open-air side (**a**) (see also (**c**)). Tree height was measured using a measuring pole (**b**) and direction was confirmed with a field map and compass. (**c**,**d**) Conceptual explanations of direction (**c**) and height (**d**). Samples from Clones 1 and 8 were collected from two different directions (Clone 1: N and W, Clone 8: E and S). Clones 2, 3, 6, and 7 were sampled from only one direction (Clones 2 and 3: N, Clones 6 and 7: S). Clones 4 and 5 were not sampled because they had no open-air side. Wogon-sugi (“W”) was sampled in all directions because it was solitary in the field. (**e**,**f**) Inflorescences from Iwafune-5 (**e**) and Higashikubiki-3 (**f**). Several male strobili develop from a single inflorescence. Arrows indicate male strobili (**e**, right; see also Figure 2a–c). Scale bar = 1 cm.

**Figure 2 plants-10-00856-f002:**
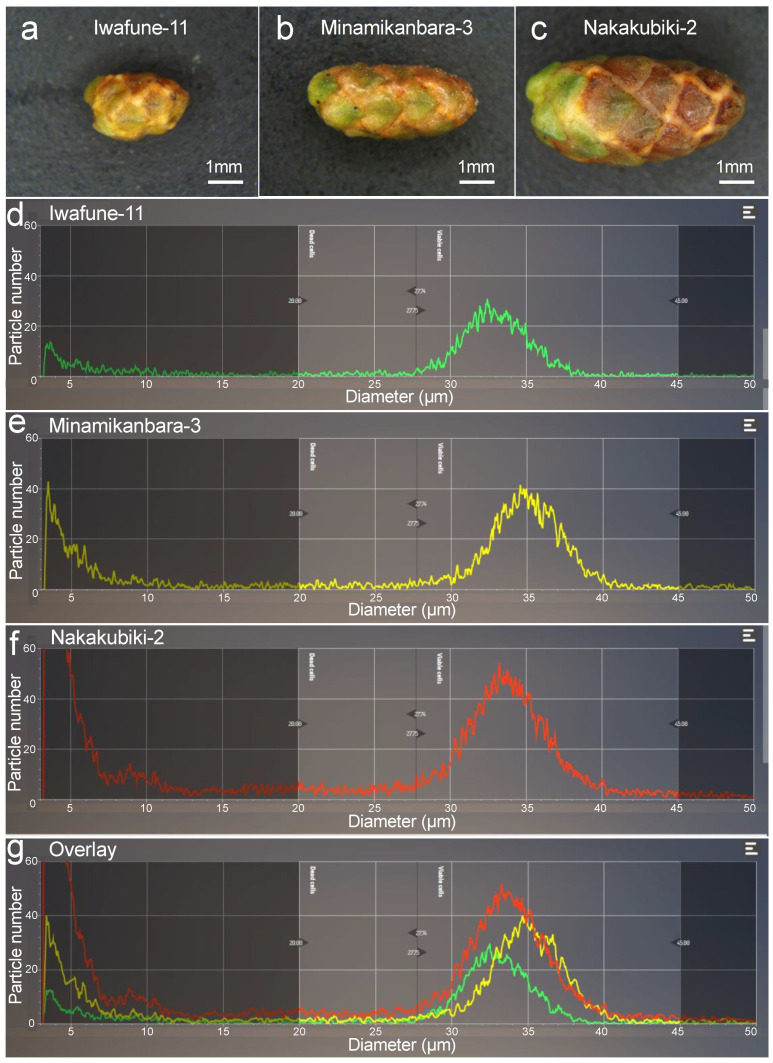
Variation in male strobili, pollen number, and pollen size. Representative samples from Iwafune-11 (**a**), Minamikanbara-3 (**b**), and Nakakubiki-2 (**c**). Scale bars = 1 mm. Particle distribution at Iwafune-11 (**d**), Minamikanbara-3 (**e**), and Nakakubiki-2 (**f**), and overlay view (**g**). The *x*-axis represents particle diameter (μm) and the *y*-axis represents particle number.

**Figure 3 plants-10-00856-f003:**
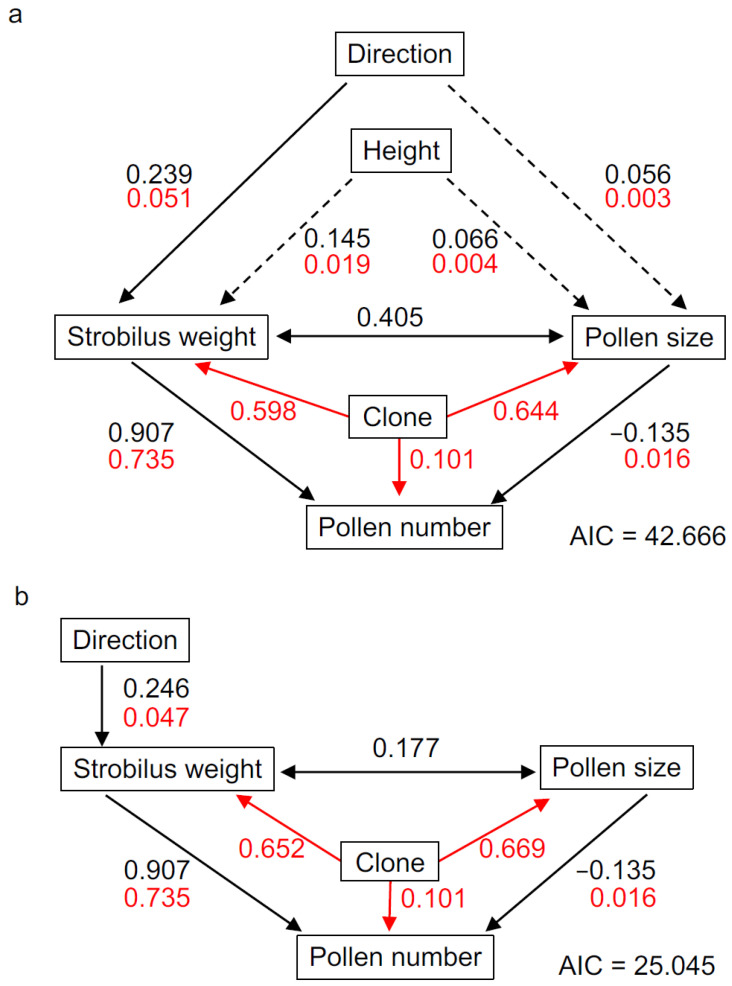
Piecewise Structural Equation Modeling (pSEM) of factors affecting pollen number. Hypothetical model (**a**) with six factors (pollen number, pollen size, male strobilus weight, direction, height, and clone) and the best model (**b**) with five factors (pollen number, pollen size, male strobilus weight, direction, and clone). Solid black arrows indicate a significant effect (*p* < 0.001), and dashed arrows indicate no significant effect. Double-headed arrows indicate that two factors are related but the direction of the effect is unclear. Black numbers indicate standardized effect size, and positive and negative numbers represent positive and negative effects, respectively. Red numbers indicate the contribution ratio (0–1) of each factor in the model. Note that numbers of the same color are comparable, whereas those of differing colors are not (see also Section 2.4).

**Figure 4 plants-10-00856-f004:**
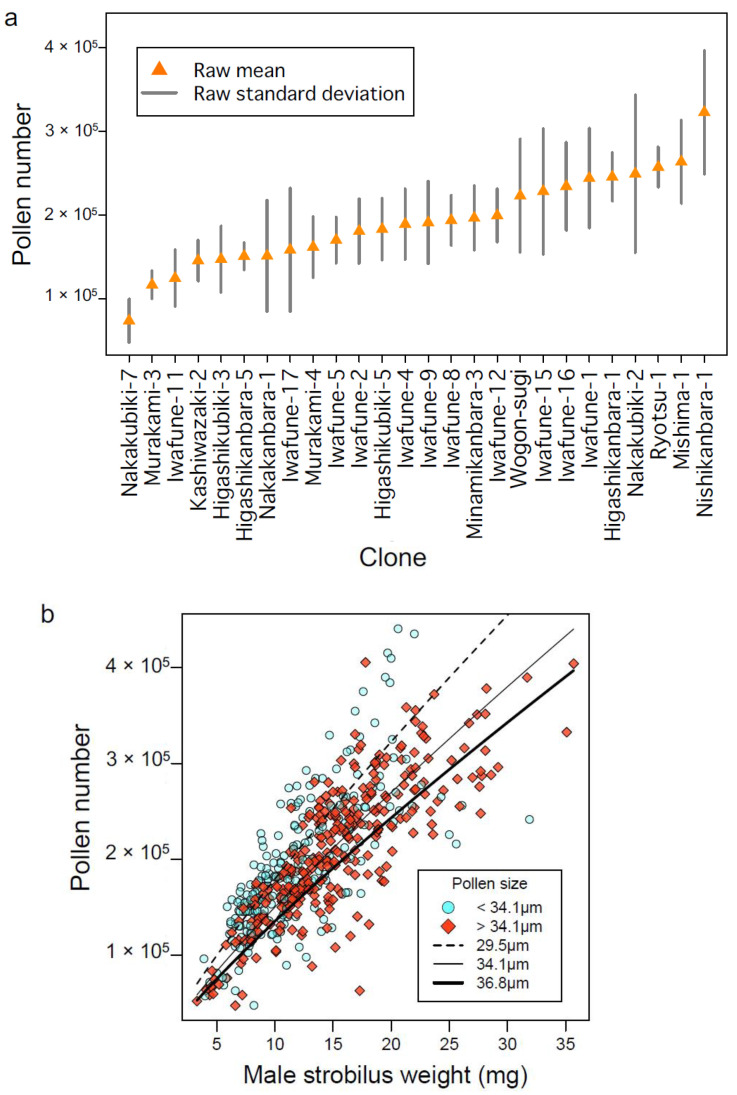
Factors affecting pollen number. Distribution of pollen number among clones (**a**). This plot was made using raw data, and thus reflects different environmental conditions (see Section 3.6). Scatterplot of pollen number and male strobilus weight from all samples (**b**). Samples with smaller pollen grains (< 34.1 μm) are indicated by light blue circles and those with larger grains (> 34.1 μm) are indicated by orange diamonds. Trend lines indicate predicted pollen number as a function of grain size: small (29.5 μm; dashed line), middle (34.1 μm; solid line), and large (36.8 μm; bold line).

**Figure 5 plants-10-00856-f005:**
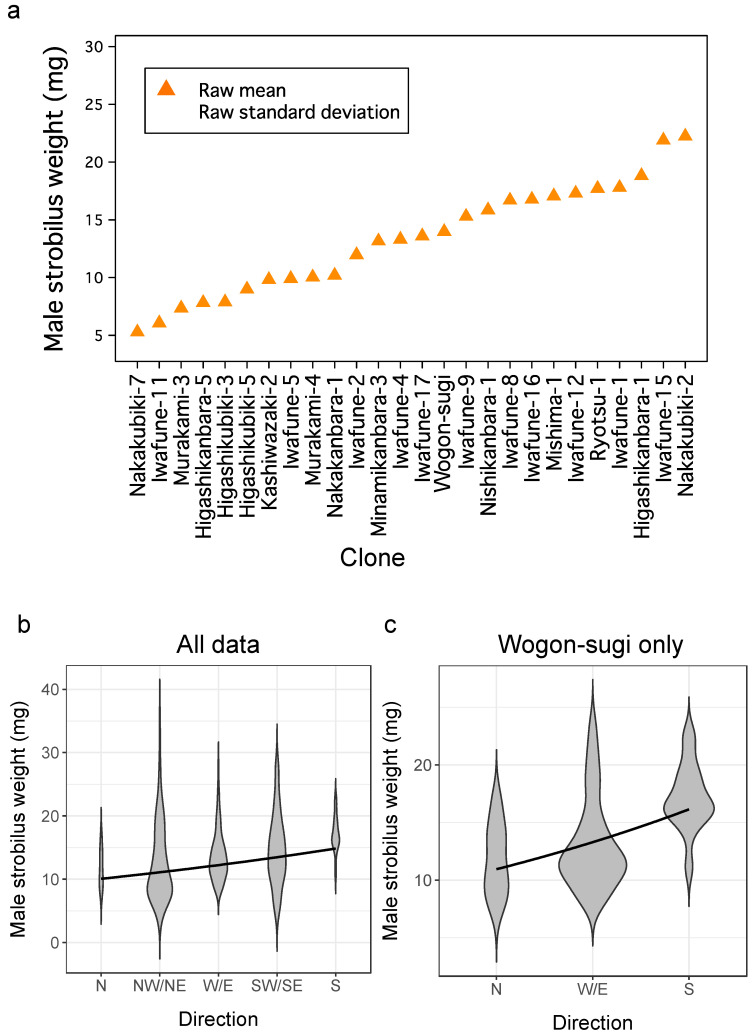
Factors affecting male strobilus weight. The distribution of male strobilus weight among clones (**a**). (**b**,**c**) Violin plots of male strobilus weight and direction with the full dataset (*n* = 523; (**b**)) and Wogon-sugi (*n* = 101; (**c**)). Plots and the trendline show male strobilus weight, which was greater on the south side. The trendline was obtained from the linear mixed effect model (LMM) included in the best model (Figure 3b).

**Figure 6 plants-10-00856-f006:**
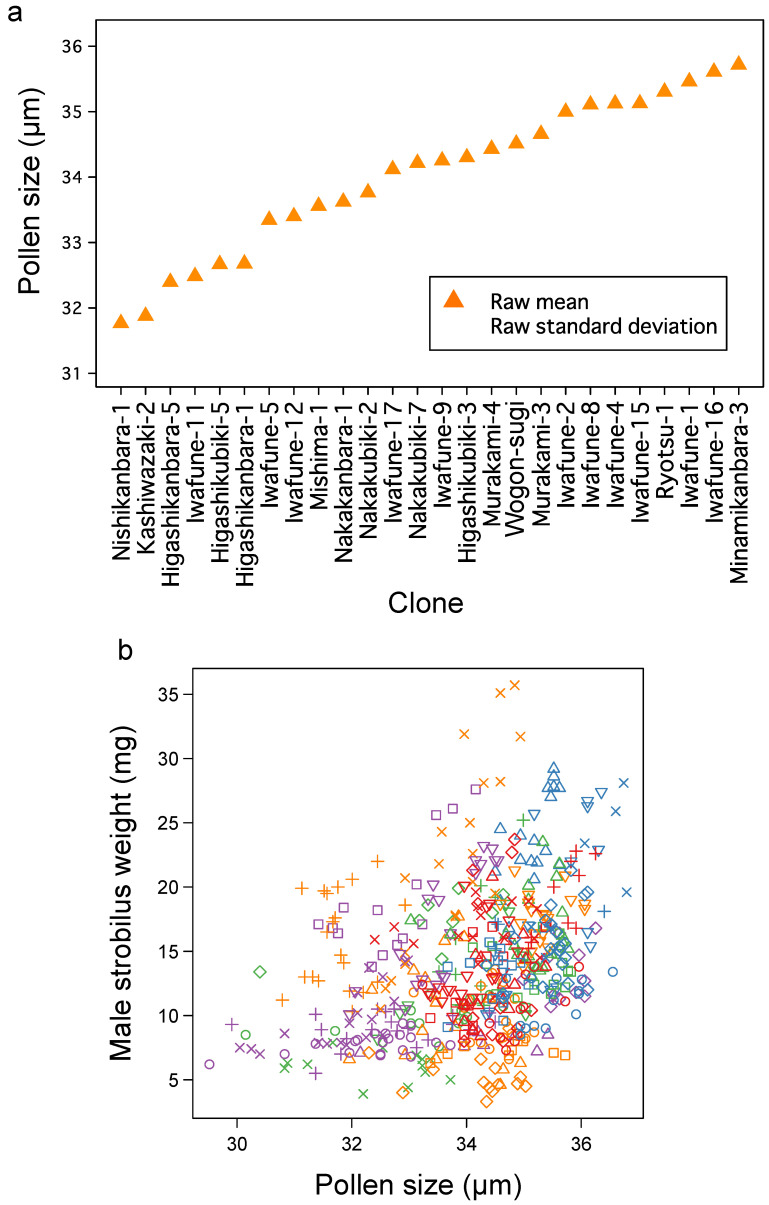
Factors affecting pollen size. Distribution of pollen size among clones: scatter plot of pollen number and pollen size (**a**). The relationship between pollen size, male strobilus weight, and clone (**b**). Each symbol indicates a clone (see Table 1).

**Figure 7 plants-10-00856-f007:**
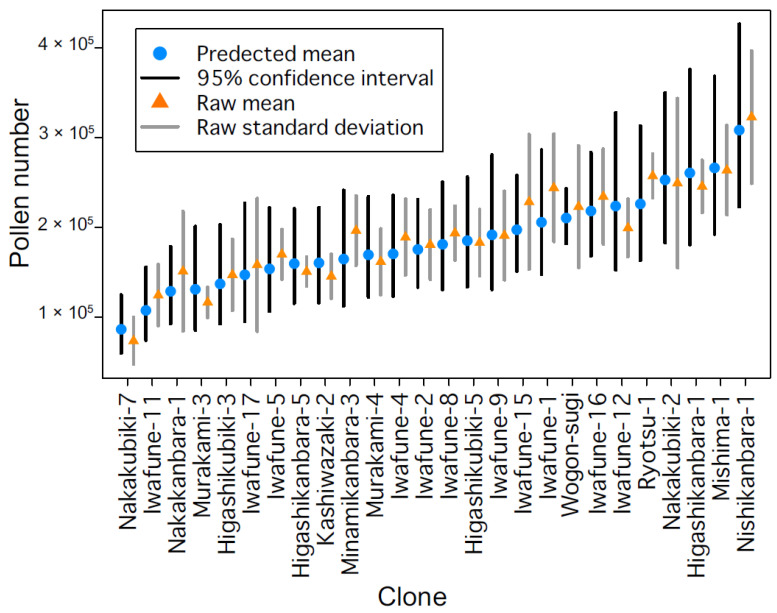
Comparison of corrected pollen numbers. Pollen numbers per clone based on raw (orange) and predicted (blue) data. Predicted data were corrected for the effects of sampling direction using the LMMs included in the best model (Figure 3b).

**Table 1 plants-10-00856-t001:** Summary of male strobilus samples in this study.

Clone Name	Direction *	Height **	Inflorescence No. ^†^	Male Strobilus No. ^‡^	Symbol ^§^
Wogon-sugi	N	>5 m	2	10	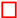
Wogon-sugi	N	<5 m	4	12	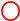
Wogon-sugi	E	>5 m	4	19	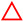
Wogon-sugi	S	>5 m	3	13	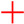 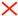
Wogon-sugi	S	<5 m	4	16
Wogon-sugi	W	>5 m	3	11	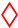
Wogon-sugi	W	<5 m	4	20	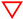
Iwafune-2	W	>5 m	3	13	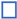
Iwafune-2	W	<5 m	3	14	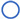
Iwafune-15	SE	>5 m	4	18	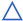
Iwafune-15	NW	>5 m	3	6	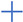 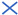
Iwafune-16	W	>5 m	3	8
Iwafune-16	W	<5 m	4	20	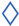
Iwafune-1	SE	>5 m	4	16	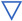
Iwafune-4	SW	<5m	4	20	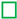
Iwafune-5	SE	<5 m	3	14	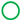
Iwafune-8	SW	>5 m	4	20	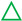
Iwafune-9	SE	>5 m	3	11	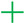 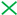
Iwafune-11	SE	<5 m	3	13
Iwafune-12	NE	>5 m	3	8	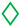
Iwafune-17	SE	>5 m	2	8	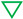
Higashikanbara-1	NW	<5 m	3	14	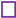
Higashikanbara-5	NW	>5 m	4	20	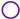
Higashikubiki-3	NE	>5 m	3	7	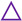
Higashikubiki-5	NW	<5 m	4	20	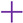 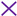 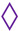
Kashiwazaki-2	NW	<5 m	4	20
Minamikanbara-3	SE	>5 m	3	11
Mishima-1	NW	>5 m	4	20	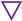
Murakami-3	NE	<5 m	2	10	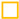
Murakami-4	NW	<5 m	4	20	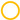
Nakakanbara-1	SE	<5 m	4	18	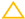
Nishikanbara-1	NW	>5 m	4	20	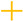 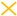
Nakakubiki-2	NW	>5 m	4	20
Nakakubiki-7	NE	<5 m	3	14	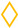
Ryotsu-1	SW	>5 m	4	19	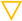
		Total	120	523	

* Direction of collected male strobilus sample (i.e., Direction of open-air side, Figure 1a,c).; ** Height of collected strobilus sample (Figure 1b,d).; ^†^ Collected inflorescence number.; ^‡^ Collected inflorescence number.; ^§^ The symbol corresponding to Figure 6b and Appendix A.

## Data Availability

The datasets used and/or analyzed in this study are available from the corresponding author upon reasonable request.

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
