# Peer review of "Factors Affecting the Number of Pollen Grains per Male Strobilus in Japanese Cedar (Cryptomeria japonica)"

_plants, 2021, doi:10.3390/plants10050856_

Round 1

Reviewer 1 Report

I am interested in the results of the study entitled ‘Factors affecting the number of pollen grains per male strobilus (flower) in Japanese cedar (Cryptomeria japonica)’. I would like to inform you that this paper accepts be printed for the following reasons. The model for predicting the number of pollen in trees is considered to be a very progressive result.

Experimental design and research results are well expressed to suit the purpose. It is necessary to reconfirm that the form of the table and figure conforms to the regulations of the Journal.

Author Response

#Reviewer 1

>I am interested in the results of the study entitled ‘Factors affecting the number of pollen grains per male strobilus (flower) in Japanese cedar (Cryptomeria japonica)’. I would like to inform you that this paper accepts be printed for the following reasons. The model for predicting the number of pollen in trees is considered to be a very progressive result.

>Experimental design and research results are well expressed to suit the purpose. It is necessary to reconfirm that the form of the table and figure conforms to the regulations of the Journal.

We thank the reviewer for this positive assessment of our work. We will carefully confirm (and modify, if necessary) tables and figures if we can publish in Plants.

Reviewer 2 Report

The multifactor analysis revealed the causal relationship between the factors.

The authors assume that genetic factors affect the clone, weight and size of pollen grains, and what exactly is given in the literature about this?

Conducting a genetic analysis in the study of pollen grains would be relevant. For example, what genes are involved in the formation and number of pollen grains?

Author Response

>The multifactor analysis revealed the causal relationship between the factors.

>The authors assume that genetic factors affect the clone, weight and size of

pollen grains, and what exactly is given in the literature about this?

>Conducting a genetic analysis in the study of pollen grains would be relevant. >For example, what genes are involved in the formation and number of pollen grains?

We thank the reviewer’s comment. We agree with the reviewer’s comment about the importance of genetic analysis and gene involved in the pollen number, we added recent studies for the genetic analysis related to the pollen number in section 3.6 as follows (lines 385-399),

"Pollen grain number varies in several plant species and the locus or gene controlling pollen number was discovered recently. Nguyen et al. investigated rye chromosome addition lines in wheat and found that the 4R addition line had 33% more pollen, which might have applications in breeding hybrid wheat [43]. Recently, the gene controlling pollen numbers was identified in Arabidopsis thaliana [44]: the pollen number varied around 4-fold among 144 Arabidopsis thaliana accessions (approximately 2000–8000 pollen grains per flower). Functional analysis revealed REDUCED POLLEN NUMBER 1 (RDP1) as the gene controlling pollen number, and an evolutional analysis revealed that RDP1 is related to the recent pollen reduction in selfing A. thaliana. Although Japanese cedar has much larger genome (10.8 GB) than A. thaliana, a recent genetic study identified the male sterility gene in Japanese cedar [45,46]. Combination with DNA marker analysis and efficient pollen counting method may contribute to revealing pollen number controlling locus (or gene) in the future."

Reviewer 3 Report

Dear Editor,

Thank you so much for choosing me as a reviewer of the manuscript ID plants-1181437 “Factors affecting the number of pollen grains per male strobilus (flower) in Japanese cedar (Cryptomeria japonica)”. I hope that my comments will help Authors to improve their manuscript.

Detailed remarks concerning manuscript:

Please add the number of lines.

Title of the manuscript.

There is no need to use the word “flower” in the title of the manuscript. It should be removed.

Abstract:

Please give not only the purpose of the report, but also the scientific hypothesis of the studies. Include the answer for the question stated as scientific hypothesis. Please present the clear conclusions for the results obtained in this studies.

Key words:

It is not recommended to give as key words the same words or phrases used in the title of the manuscript. You may also use as key word the word “flower” removed from the title of the manuscript.

Introduction

1st paragraph

Cited references no 2-7, 27 refer to the evolution. Please skillfully combine it with the topic of the work.

Please refer to Cryptomeria japonica, not to taxonomically unrelated distant species: Actinidia deliciosa, Oryza perennis, Pyrus pyrifolia, Triticum aestivum [7-12].

"Allergic reaction called pollinosis", the cited items do not contribute to scientific knowledge [13, 14].

2nd paragraph

With reference to the topic of the work, please describe the occurrence of Cryptomeria japonica in Japan.

The issue of allergy caused by Cryptomeria japonica and other species of the family Cupressaceae requires update data. Please include the latest literature data. Citations 14-17 including the years 2007, 2005, 200, 2014.

  1. Data on allergies include the years 1998 and 2008 [20], please include new references.

Some information should be included into the methodology section (the same concerns the other information from introduction) e.g. „The total number of pollen grains produced by a single tree is the product of the number of pollen grains per male strobilus (male flower) and the number of male strobili per tree.”

3rd paragraph

Citation no 22 is in Chinese, only an English abstract is available, therefore the analysis of the original text is impossible.

4th paragraph

Explain some phrases, for example, "hypothesized causal", "relationships between factors" "complex network structures between interrelated factors".

The authors citing five references do not present but signal information [27] [28] [29] [30] [31].

Generally

Complete the research theses related to the topic of work.

Clearly define and justify the purpose of the work

The Introduction section should be divided into subsections describing the background of the studies.

Materials and Methods

page 3, 4

The place of plant material collection should be précised, please provide navigational data

In which stage or phase the “ Male strobili” collected?

“Samples were collected from at least two inflorescences, and pollen number was counted from at least six samples from each sample or sampling condition” This sentence needs explanations. What it means “six samples from each sample or sampling condition”?

Please complete the description of the table

Page 6, 7.

On the photos a, b the bar value should be given, while on the photos d-g provide the explanatory symbols of the X and Y axes.

Page 9.

Please give the information concerning statistical analysis.

Results and Discussion

Page 9-18,

3.1.

Please remove repetitive information, e.g. "523 male strobili collected from 26 clones" - page 3; 4; 9 (it is impossible to mention all of them, the same concerns the whole section).

Some of the information concerns methodology.

Results should be presented clearly and precisely

Expand the discussion in points 3.2-3.7

3.2.

Authors should distinguish the information included in "Material and methods" section.

In the "Results and Discussion" section the specific obtained results should be presented and the scientific achievements of other authors should be referred to results (it concerns to the whole section).

Reference [32] does not represent a discussion with the results.

3.3.

The chart type can be seen “Figure 4a shows a bar plot of pollen number by clone.”

Information: “This plot indicates”, “Note that this plot did not consider”, “Figure 4b shows” – should be removed.

3.4.

The citation [38, 32] did not bring new information and arguments to discuss.

“as the south side of trees in Japan receive more sunlight than the north side [38]” “This is supported by previous findings that showed that male strobilus growth is affected by light receipt [32].” – It is obvious.

Figure a - complete the marking of the X axis (similarly in the further results)

3.5.

The discussion of results should not be based on the statement. „The number of male strobili is positively controlled by sunlight [32].”

3.6.

The reader expects clear and concrete results.

Figure a, complete the marks for the X axis. The units “µ” should not be italicized.

“In general the Results and discussion section needs a major changes. It should be rewritten.

In fact Authors did not discuss their results.”

Please indicate the group of people for which the research may be useful.

Please give clear direction for the future studies.

The figures are not marked numerically what make difficult to find them. The reader may only guess it.

Conclusions

Please do not repeat the same information twice.

“The total pollen number from a single tree …pollen grains per male strobilus.” Page 1; 2; 19

Based on the obtained results, give specific conclusions for the purpose of the studies and research theses.

Figures

The description and explanations of the figures are too long should be prepared according to the guides for authors. However, all the figures, just like the tables, should be clear for the reader without referring to the text of the manuscript and all needed information should be included. Please check the compliance of marks. Please give the numerical values for the bars.

Author Contributions

The sentence “All authors have read and agreed to the published version of the manuscript” should not be bolded. Remove information “add later”.

Reference list

Reference list should be prepared strictly to the guides for authors [for example reference no 13, 23, 35, 37, 39, 40, 52, 57, 59, 74]. There are many editorial mistakes in the reference list. It is not possible to indicate all of them. For example: some of the journals titles are abbreviated but the other are not.

Standardize the record:

- "doi:" e.g. 1, 2

- books e.g. 1, 13, 18,18, 19, 25, 34

- journal abbreviations, e.g. 8, 11, 21, 22, 24, 32, 38

Instead of old references from 1987, 1992, 1997, new ones can be cited; moreover it is difficult to check the original in Japanese.

Citations from 2015-2021 includes 31.6% of all references, but they should include at least 50% of all items.

I strongly recommend to submit the paper after needed changes.

Author Response

Dear Reviewer 3

First of all, we apologize to the reviewer for our mistaken formatting (for example, lacking line numbers, no Figure numbers, many mistakes of reference formatting). We carefully checked the formatting and fixed them.

We agree with the reviewer’s comment about our few discussions. We added discussion sentences to sections 3.2-3.6, (especially we largely added discussion sentences in section 3.6; lines 340-399).

Our revised manuscript was checked and edited by the English correction service (Textcheck: http://www.textcheck.com/en/text/page/index ). 

Detailed point-by-point responses to the reviewer’s comments are provided attached file.

Round 2

Reviewer 3 Report

Detailed remarks:

Figure 1e, f bars should be placed in the left bootom corner. Now the manuscript may be published.